# Bioconversion of Agricultural Wastes into a Value-Added Product: Straw of Norwegian Grains Composted with Dairy Manure Food Waste Digestate in Mushroom Cultivation

**Agnieszka Jasinska** [1,2,*], **Ewelina Wojciechowska** [3], **Ketil Stoknes** [1] **and Michał Roszak** [4]

1    Lindum AS, Lerpeveien 155, 3036 Drammen, Norway; ketil.stoknes@lindum.no
2    Department of Vegetable Crops, Poznan University of Life Sciences, ul. J.H. Dabrowskiego 159, 60-594 Poznan, Poland
3    Norwegian Institute of Bioeconomy Research (NIBIO), Holtveien 66, 9269 Tromsø, Norway; ewelina.wojciechowska@nibio.no
4    Soppas, Nedre Keisemark 1E, 3183 Horten, Norway; loszak93@gmail.com
*    Correspondence: agnieszka.jasinska@lindum.no or agnieszka.jasinska@up.poznan.pl

**Abstract:** Commercial mushroom production is based on composted locally available agro-industrial wastes rich in carbon and nitrogen such as wheat straw supplemented with chicken manure. Either component can be replaced by other kinds of grain straw: barley, oat, or a mixture of different straw types and combined with diary manure—food waste digestate after anaerobic biogas digestion. Original, unseparated liquid digestate is nutritious, rich in nitrogen and organic matter. This research aimed to investigate the effect of digestate and different straw ratios on the composting process and productivity and their consequent effect on mushroom cultivation parameters of *Agaricus subrufescens*. All investigated experimental mushroom compost (EMC) types worked well during the composting process, reaching the desired moisture of 65–75%, N content of 1.43–1.93%, and a C/N ratio ranging from 21.5 to 29.1, supporting growth of mycelium and producing mushrooms. Supplementation with barley straw resulted in better EMC structure with the highest yield and biological efficiency (BE) (157.9 g kg$^{-1}$; 64%), whereas oat addition gave the lowest yield and BE (88.6 g kg$^{-1}$ and 38%). Precociousness (yield at mid-cycle of the crop development) was higher for oat substrates (68.9%), while earliness (days to harvest from casing) was lower for barley EMC.

**Keywords:** *Agaricus subrufescens*; composting process; combined digestion; sustainable mushroom cultivation; agricultural waste reuse; cultivation substrate optimization; fungi; barley straw; oat straw; mushroom composts


## 1. Introduction

New agricultural wastes appear with the growing awareness and importance of the circular bioeconomy and a need for reuse and recycling of waste. Additionally, with the industrialization and a growing urbanization of the world, the amounts of food waste and improper waste management has been increasing. One-fourth of the produced calories in the world are lost or wasted [1,2]. Collaboration between the food production and waste management sectors is especially important to keep nutrients and organic matter in productive loops rather than discarding them as waste through landfilling or incineration. Considerable amounts of carbohydrate, protein, and fat can be found in food waste. What is more, this material has high moisture content, therefore it could be used in anaerobic digestion (AD), chemical hydrolysis, or aerobic composting [3–7]. In regions where the infrastructure is already there, combined feedstock can increase biogas production [8,9]. Anaerobic co-digestion of diary manure and food waste is practiced in The Magic Factory in Sem, Norway. AD leaves an effluent called 'digestate'. This nitrogen- and organic-matter-rich product has been investigated as an ingredient in growing substrates suitable for

mushroom cultivation [9–16]. In this way, mushroom cultivation naturally combines waste management with food production [17,18], developing new markets for waste materials after AD.

For the referenced studies, *Agaricus* species were chosen, since the aim was to utilize substrates based on food waste, which have a high protein content (lower C/N ratio). A high nitrogen content is beneficial for the button mushroom so long as ammonia is well managed at the end of the second stage of mushroom compost preparation [19]. Besides a source of nitrogen, mushrooms require carbon for growing and developing into mature specimens. The spectrum of substrates which can be used as growing substrate for mushrooms is numerous [20,21]. Most research on mushroom compost for *Agaricus* ssp. mushrooms was performed with wheat straw as the main component. Norwegian grain production is estimated at 1.1 to 1.3 million tons per year and is mainly barley (570,000 tones), wheat (450,000 tones), oat (230,000 tones), and rye (50,000 tones) [22]. Thus, there is a potential to investigate barley and oat in addition to wheat straw for mushroom cultivation. However, Norwegian mushroom consumption reaches 7000 tons yearly of imported mushrooms, mainly white and brown button mushrooms; there is no domestic commercial production. Norway is now promoting circular bioeconomy, short supply chains, locally sourced and produced products; hence, it is highly appropriate to investigate commercial button mushroom production using local resources. Therefore, this study aims to examine the utilization of straw of local grains and digestate from combined diary manure–food waste AD for cultivation of almond button mushroom *Agaricus subrufescens*, a lesser known cultivated *Agaricus* species in the European market. This mushroom prefers warm temperatures 23–27 °C, is tolerant of lower relative humidity in the cultivation chamber, and is pest and disease resistance [23]. Called the almond mushroom, for it contains benzaldehyde and benzoic acid, giving it a special almond-like smell and taste, which makes it extraordinary culinary gourmet food [24,25]. The species is considered a medicinal mushroom, containing bioactive polysaccharides and protein complexes (PSPC) which have been shown to function as potent antioxidant, antitumor, and anticancer agents [26–28].

Moreover, this approach will be in line with closing material, energy, and nutrient loops through "reducing, actively reusing, recycling and recovering materials" [29–31].

The following research addressed the following objectives:

1. Investigate the effectiveness of using original (not separated into liquid and solid fraction) combined food waste—diary manure digestate, hereafter called 'original digestate' (OD), and straw of Norwegian grains in Phase I and II composting.
2. Examine a range of different straw ratios to determine suitable mushroom compost composition for *A. subrufescens* mushroom cultivation.
3. Determine the influence of straw type and OD on *A. subrufescens* productivity parameters: yield, biological efficiency (BE), and dry matter (DM) of mushroom fruiting bodies and on mushroom cultivation parameters: earliness, precociousness, and number of mushrooms.

## 2. Materials and Methods

### 2.1. Fungal Strains

Spawn of Agaricus subrufescens Peck (M 7700) was purchased from the mushroom spawn laboratory MYCELIA, Deinze, Belgium.

### 2.2. Mushroom Compost and Casing Material

Experimental mushroom composts (EMC) consisted of corn straw of three species; wheat (*Triticum aestivum*) spring variety Mirakel, barley (*Hordeum vulgare*) late variety Thermus, and oat (*Avena sativa*) late variety Hurdal, grown in Vestfold and Telemark county, Norway, chopped at lengths of 5 cm; source-separated fresh food waste (80%) and diary manure (20%) original digestate (OD) from anaerobic digestion treatment processes were combined—(source: municipal biogas plant The Magic Factory, Sem, Norway), chicken

manure (source: Felleskjøpet, Drammen, Norway), hot compost (Lindum AS—outdoors active composting windrows of garden waste), gypsum (source: fraction from recycled plaster boards from Gips-recycling Norge AS, Norway), casing material—Norwegian Black Peat (Holmen Torv). Physical and chemical compositions of the materials used in these experiments are given in Table 1.

**Table 1.** Raw materials used.

| Material | pH | DM (%) | EC (mS cm$^{-1}$) | Ash (%) | N (% DM) |
|---|---|---|---|---|---|
| Wheat straw used in Experiment 1 | - | 95.9 | - | 5.4 | 0.60 (0.01) |
| Wheat straw used in Experiment 2 | - | 96.0 | - | 7.9 | 0.30 (0.02) |
| Barley straw | - | 95.8 | - | 8.5 | 0.25 (0.02) |
| Oat straw | - | 95.8 | - | 7.6 | 0.13 (0.003) |
| Digestate Experiment 1 | 8.1–8.3 | 4.9–5.4 | 20.0 | 28.1–30.05 | 0.60 (0.01) |
| Digestate Experiment 2a | 8.0 | 5.0 | 22.0 | 26.3 | 0.98 (0.04) |
| Digestate Experiment 2b | 8.4 | 4.6 | 20.0 | 27.1 | 0.96 (0.04) |
| Chicken manure | 5.6 | 45.0 | 13.4 | 10.0 | 0.14 (0.01) |
| Hot compost | 7.0 | 34.0 | 0.4 | 18.7 | 0.12 (0.005) |
| Norwegian dark peat for casing | 7.9 | 24.0 | 1.3 | 39.6 | |

Values in parentheses are the standard deviations, DM (dry matter), EC (electrical conductivity).

### 2.3. Experiment Set-Up and Measurements

Two successive experiments were performed at the climate-controlled mushroom growing chamber of the R&D Department of Lindum AS. Composting was medium scale (20–40 kg of EMC per batch) in reactors mimicking commercial conditions.

#### 2.3.1. Experiment 1—Use of Original Digestate (OD) in *A. subrufescens* Cultivation

The purpose was to investigate digestate of food waste and diary manure as an ingredient in the composting process. Four batches were made representing combinations of materials and ratios, the amounts were estimated by dry matter of substrates (all experimental composts made in the study are individually listed in Table 2).

**Table 2.** Experimental mushroom composts—at make-up and during Phase I.

| EMC | Wheat Straw (% DM) | Oat Straw (% DM) | Barley Straw (% DM) | Digestate (% DM) | Chicken Manure (% DM) | Hot Compost (% DM) | Gypsum (% DM) | Digestate Water (g kg$^{-1}$) | Max. Temperature (°C) in Phase I | DM (%) in Phase I |
|---|---|---|---|---|---|---|---|---|---|---|
| | | | | Experiment 1 | | | | | | |
| WD1 | 82.0 | - | - | 10.0 | 3.1 | - | 4.6 | 644 | 80 | 33.4 |
| WD2 | 84.0 | - | - | 7.0 | 4.0 | - | 5.0 | 556 | 74 | 39.4 |
| WD3 | 79.8 | - | - | 13.8 | 2.1 | - | 4.4 | 700 | 72 | 28.8 |
| WD4 | 80.7 | - | - | 13.0 | 2.4 | - | 4.1 | 705 | 72 | 27.9 |
| | | | | Experiment 2a | | | | | | |
| WD | 80.0 | - | - | 12.5 | 3.8 | - | 3.9 | 695 | 64 | 28.9 |
| O4WD | 39.7 | 39.7 | - | 12.4 | 3.7 | - | 4.6 | 675 | 81 | 29.0 |
| B4WD | 39.7 | - | 39.7 | 12.4 | 3.7 | - | 4.6 | 675 | 81 | 29.0 |
| | | | | Experiment 2b | | | | | | |
| O25WD | 60.7 | 24.2 | - | 10.2 | - | 2.7 | 2.1 | 572 | 70 | 28.4 |
| B25WD | 58.5 | - | 25.0 | 11.2 | - | 3.0 | 2.3 | 590 | 73 | 26.5 |
| BOWD | 33.6 | 16.8 | 33.4 | 10.9 | - | 3.0 | 2.4 | 595 | 71 | 27.7 |
| B6WD | 25.2 | - | 58.6 | 10.9 | - | 3.0 | 2.4 | 593 | 77 | 27.7 |

DM—dry matter, experimental mushroom composts (EMC): WD—wheat straw–digestate (1,2,3,4-consecutive numbers of EMC); OWD—oat–wheat straw–digestate; BWD—barley–wheat straw digestate; BOWD—oat–barley–wheat straw digestate; the numbers in exp 2a and 2b means the % of oat or barley straw in substrate).

### 2.3.2. Experiment 2a and b: Use of Optimal Corn Straw Combination in *A. subrufescens* Cultivation

The purpose was to investigate the three native straw types (wheat, oat, and barley) and original digestate as an ingredient in the composting process. Seven batches were made, representing combinations of materials and ratios by dry matter. Additionally, two types of compost processing triggers were used: chicken manure, in conventional mushroom cultivation mushroom compost (exp. 2a), and hot compost from active composting windrows of garden compost (2b), all experimental mushroom composts made in the study are individually listed in Table 2.

### 2.4. Mixing and Composting Procedures

Mixing was performed alternately by hand, hayfork, and by means of a hand-held cement mixer (Elektromix ZY-HM-140; Yongkang Well-King Industry and Trade Co. Ltd., Jinhua, Zhejiang, China) in a 500 L plastic container.

The composts' compositions were based on dry matter (DM) of substrates before processing, and were evened to obtain ±30% of DM of substrate. The digestate from AD was hygenized (the AD plant uses a 70 °C for 1 h, then digestion on 40 °C for 30 days) before it was used in the original form and added to Phase I substrate. Chicken manure in exp.1 and 2a, and hot compost in exp 2b was added to trigger the composting process and gypsum to maintain suitable compost structure and stable pH of the outcoming substrate around 7.0–7.3. Independent samples of straw, digestate, chicken manure, and hot compost (different batches collected at different times) were collected for each round of experiments.

### 2.5. Compost Preparation

### 2.5.1. Phase I—The Composting Process

Mushroom growing substrate was composted in rotating 270 L composter drums (JK 270; Joraform, Mjölby, Sweden) for 4 days, until volume loss due to heat generation. Later, substrate was mixed and moved to insulated containers with controlled air flow through the substrate (to obtain similar conditions as in commercial bulk systems; 70–80 °C and 6–9% $O_2$ $v/v$). The $O_2$ level was measured three times a day with portable gas analyzer (GA5000, GeoTech, QED Environmental Systems Ltd., Coventry, UK). The substrate was turned again twice at an interval of 3 days, giving total of 10 days for Phase I.

### 2.5.2. Phase II—Pasteurization

The mushroom substrate was moved into a miniature Phase II tunnel, where temperature was slowly increased to 60 °C and then maintained for 6 h of pasteurization. During conditioning, temperature was decreased to 55 °C for another 6 h and next to 50 °C, until ammonia ($NH_3$ gas) had dissipated (below 10 ppm). The $NH_3$ gas level was measured with Dräger accuro pump for gas detection tubes ammonia 2/a 2–30 ppm (Dräger Inc., Huston, TX, USA). Composition of the compost after Phase II (at inoculation) used in both cultivation experiments are listed in Table 3. More detailed description of Phase I and Phase II processes are described by Stoknes [12].

**Table 3.** Composition of the experimental mushroom compost after Phase II (at inoculation).

| EMC | DM % | N (%) | C/N Ratio | Bags (Repetitions) |
|---|---|---|---|---|
| | | Experiment 1 | | |
| WD1 | 35.9 | 1.62 | 25.7 | 7 |
| WD2 | 29.5 | 1.41 | 29.7 | 7 |
| WD3 | 21.9 | 1.80 | 22.7 | 14 |
| WD4 | 24.6 | 1.87 | 22.1 | 10 |

**Table 3.** *Cont.*

| EMC | DM % | N (%) | C/N Ratio | Bags (Repetitions) |
|---|---|---|---|---|
| | | Experiment 2a | | |
| WD | 25.3 | 1.88 | 21.9 | 9 |
| O4WD | 23.1 | 1.89 | 21.9 | 6 |
| B4WD | 25.8 | 1.89 | 21.9 | 5 |
| | | Experiment 2b | | |
| O25WD | 26.8 | 1.69 | 25.2 | 5 |
| B25WD | 24.8 | 1.80 | 23.5 | 5 |
| BOWD | 25.7 | 1.79 | 24.0 | 9 |
| B6WD | 24.1 | 1.80 | 24.1 | 9 |

DM—dry matter, RH—relative humidity of experimental mushroom composts; experimental mushroom composts (EMC): WD—wheat straw–digestate (1,2,3,4-consecutive numbers of EMC); OWD—oat–wheat straw–digestate; BWD—barley–wheat straw–digestate; BOWD—oat–barley–wheat straw digestate; the numbers in exp 2a and 2b means the % of oat or barley straw in substrate).

### 2.6. Mushroom Cultivation and Cropping Procedures

#### 2.6.1. Bag Cultivation

During cultivation experiments, 50-micron polypropylene, autoclavable bags of capacity of 7 L, flat dimensions of 40 cm wide × 51 cm high (type PP50/SEU4/V40-51, SacO$_2$ Microsac; Deinze, Belgium), with four linear ventilation filters were used. Bags were filled with approximately 3 kg of pasteurized EMC and left to cool to room temperature (22 °C). The EMC was then inoculated with granular spawn on wheat grain, applied at an amount of 3% of fresh substrate weight (90 g). The bags were sealed and shaken by hand until the spawn was evenly distributed.

#### 2.6.2. Incubation, Casing, Pinning, and Fructification

Bags were randomized and incubated at 25 °C in a dark room with internal air circulation, passive ventilation, and no humidification. After the spawn overgrew the EMC, bags were opened and a 5–7 cm layer of casing (approx. 700 g) was applied and left for another week, covered by perforated agrotextile. As casing material, Norwegian black peat (from Holmen Transport) of EC 1 mS cm$^{-1}$ addition 3 g of Ca(OH)$_2$ for 1 kg of casing to adjust the pH to approx. 7–7.5 and gypsum (120 g gypsum/40 L peat) for structure maintenance were used. Casing is crucial as it holds enough water for formation and development of basidiocarps (i.e., mushrooms). The agrotextile was taken away daily for sprinkling with water (evenly distributed over all the bags). Bags were placed in the climate-controlled mushroom growing chamber. At the end of this week, mycelium was present at the surface, which was then ruffled (scratching through the casing layer). The temperature inside the growing chamber was 25 ± 2 °C for 15 days than decreased by 5 °C for next 5 days for initiation of pining. The air humidity for fruit body development was held at 85–95%. The cultivation chambers received LED light with color temperature 6000 K. The cultivation room was ventilated so that CO$_2$ concentration did not exceed 1000 ppm.

#### 2.6.3. Harvest

Mushrooms were picked at maturity: at a closed stage (veil clearly visible, but not ruptured). The soil-covered basis of the stalk was removed by trimming and was not included when weighing. Mushrooms were counted. All mushrooms in a single bag were harvested at once. Yields were determined as the weight of the harvested mushrooms from the complete cropping period per fresh weight of substrate at inoculation.

A representative sample of each bag crop (approx. 100 g) was also weighed, dried, and kept for quality analysis.

*2.7. Analysis of Raw Materials and Experimental Mushroom Composts*

2.7.1. Sampling and Homogenization

Fresh samples (300 g) of thoroughly mixed EMC or raw materials were taken as composite samples by combining 10 random sub-samples. From this, fresh samples were taken for immediate pH and EC (electrical conductivity) measurement (repeated three times). The remaining samples were dried and weighed for dry matter determination, combined, and subsequently homogenized in a blender (ES3Xpress; Blendtec, Orem, UT, USA). This was used for ash content analysis.

pH, electrical conductivity (EC), dry matter (DM) and ash content were determined immediately after sampling. Minimum 10 g EMC/substrate were dried to constant weight of the sample in laboratory dryer at 105 °C in duplicate to determine dry matter. Thereafter, the dried material was ignited in the muffle furnace at 550 °C for 2 h to determine volatile solids/ash content. Approximate OM or volatile solids was calculated as (% OM = 100% DM − % ash content). DM and ash are given as the mean of the two samples. Calculations: DM (%) = Dried material (g)/Fresh material (g). Ash (%) = Burnt material (g)/Dried material (g).

A common method of measuring soil pH/EC was used. EC and pH measuring was performed by placing a pH or EC meter in mixture of a substrate and: distilled water in ratio 1:2.5. The liquid is distilled water (for active acidity). The pH/EC was measured after 30 min using a pH meter (Milwaukee 802 pH/EC/TDS meter) and EC meter (Milwaukee 802 pH/EC/TDS meter).

2.7.2. Other Analysis

Organic nitrogen (N) was analyzed using Kjeldahl-N method according to Int/NS-EN 13342:2000 by VestfoldLab, Sem, [32,33] or by EN13654-1 m by Eurofins, Moss.

Organic carbon (C) was calculated as 58% of the organic matter (Organic matter (%) = Total organic carbon (%) × 1.72).

2.7.3. Statistical Analysis

The cultivation bag was the statistical unit in terms of the response variables:

- Productivity parameters:
  - Yield = (fresh weight of mushrooms from the whole cropping period)/(fresh weight of substrate at inoculation)
  - Biological efficiency (BE) of substrates = (fresh weight of mushrooms from the whole cropping period)/(dry weight of substrate at inoculation)
  - Dry matter of mushrooms (DM) = (fresh weight of harvested mushrooms)/(dry weight of harvested mushrooms)
- Mushroom cultivation parameters
  - Earliness (E days) = number of days between the casing and the primordia formation [34,35].
  - Precociousness (P) (yield in first half of harvest time)/(yield in total harvest time) [36]. This is a specific parameter used to monitor the yield at mid-cycle of the crop development (the higher the value of *p* the better).
  - Number of mushrooms = number of mushrooms harvested during the whole cropping period

The MINITAB version 19.2 (Minitab, Coventry, UK, 2019) statistical software was used for the one-way variance analysis (ANOVA), and the Fisher least significant difference (LSD) method ($p < 0.05$) was applied for pairwise comparison of means for all the variables.

The Pearson correlation coefficient was used to measure the strength of relationship between the response variables (yield, BE, P, E, number of mushrooms, DM).

## 3. Results and Discussion

### 3.1. Mixing and Composting of Wheat Straw with Original Digestate—Experiment 1

The aim of the first experiment in this study was to investigate the effectiveness of using original digestate for cultivation and the optimization of the composting methods of *Agaricus subrufescens*.

Original digestate was easy to mix with all kinds of straw, and the formation of digestate lumps was avoided. No additional watering was needed at the make-up point because the non-separated, original digestate was wet enough (digestate water content varied from 556 to 705 g kg$^{-1}$) to evenly cover corn stalks and gave an initial substrate moisture range from 64.1% to 78.1% at the point of inoculation (Tables 2 and 3), which is suitable for *A. subrufescens* cultivation [37–39]. Substrate components, such as gypsum and composting inoculum, were first added to original digestate and mixed to ensure even distribution throughout the substrate mass. Substrates containing a mix of straws were made by mixing dry straws together first. The barley stalk was harder than wheat and oat and therefore harder to work with. Liquid digestate tended to drain through the stalks and was left in the mixing container. To avoid loss of digestate, composting drums were sealed with foil. However, oat tended to soak quickly and form lumps.

All substrates had good temperature development during the first phase of composting, reaching a maximum of 64 to 81 °C (Table 2). All substrates developed satisfactorily through Phase I and could be transferred to Phase II—pasteurization and conditioning (Table 2). By the end of Phase II, substrates in Experiment 2 a and b were more even in terms of moisture (73.2% to 76.9%, see Table 3), whereas substrates in Experiment 1 presented higher differences (64.1% to 78.1%). Savoie et al. [11] indicated that, in their research, high levels of NH$_3$ of digestate based substrate forced long composting periods, over 20 days. In our study, NH$_3$ level in all experimental substrates dropped below 10 ppm after 24 h of Phase II and could be used in cultivation experiments. The whole composting process was 12 days. The structure of the experimental mushroom compost was even in all of the four batches of Experiment 1, with the DM at inoculation ranging from 21.9% to 35.9% (Table 3).

The cultivation of *Agaricus subrufescens* is based on a technology adapted from the cultivation of *Agaricus bisporus* (button mushroom), since both mushrooms show similar behavior [40]. First, the mushroom compost must be prepared from nitrogen and carbon rich substrates, such as straw and digestate. After biogas production, the digestate is usually separated into liquid and solid fraction. Previously, *Agaricus* species has been successfully cultivated on straw and the solid fraction of digestate from food-waste [9,12], straw mixed with paper scraps and solid digestate [15,41]. The solid fraction of digestate contains between 30% and 40% of DM, and is rich in in fiber and nutrients [42]. The main problem with the solid fraction was mixing it again with water and straw to obtain an even, homogenous composting material without lumps of digestate [12]. The present study used original, nonseparated digestate based on dairy manure and source-separated food waste. To investigate the uniformity of the recipe and stability of the composting process in Experiment 1, four consecutive batches were prepared. Substrates were composed from the same batch of wheat straw, but different batches of digestate with slightly different DM content, ranging from 4.6% to 5.4% (Table 2). However, all substrates aimed to have initial dry matter content between ± 30% and 40% (Table 2), which was appropriate to achieve good yielding of *A. subrufescens* in previous studies [12,15,41].

Effect of the C/N Ratio of the Original Digestate Based Experimental Mushroom Composts on the Mushroom Yield and Biological Efficiency (BE) of the Mushroom Composts

A composting process is necessary to obtain a selective substrate for *Agaricus* mushroom cultivation [43,44]. The initial materials are processed by natural action of several different microorganisms. Composting leads to degradation of soluble sugars and the high temperature favors development of thermophilic microorganisms which contribute to this selectivity [45,46]. The initial level of nitrogen (N-initial) in the substrate is one of the most

important conditions for the development of this microbiota and for the subsequent growth of the mycelium.

In this study, the C/N ratio ranged from 21.9 up to 29.7 (Table 3) and allowed all examined substrates to develop good temperature in Phase I (Table 2). The digestate nitrogen content, ranging from 0.6% to 1% DM (Table 1) was sufficiently available for the composting process; however, Noble et al. [47] argued differently. The N-initial of examined mushroom composts ranged from 1.41% up to 1.87% for Experiment 1 and from 1.69% to 1.89% for Experiment 2 (a, b), which is assumed to be close to ideal concentration for *A. subrufescens* according to Siqueira et al. [48]. Their article stated the ideal mushroom compost nitrogen concentration for the cultivation of *A. subrufescens* is 2%. Andrade et al. [49] obtained biological efficiency (BE) of 33.63% in the cultivation of *A. subrufescens* in mushroom compost with around 1% N-initial. In our study, the BE of all the examined substrates differed, from the lowest of just BE15% and the highest of BE64% (Table 4), N-initial of 1.62% and 1.87% (Table 3) respectively in Experiment 1 and lowest of 36 and highest BE64% (Table 4) and 1.69% and 1.80% (Table 3) respectively in Experiment 2.

All experimental mushroom composts in this study supported mycelium growth and mushroom production, but BE of the mushroom composts differed greatly, improving from batch to batch (from 15% in first batch to 64 in the last batch of the experiment). The C/N ratio of the experimental mushroom composts ranged from 22.1 up to 29.7, corresponding with what Stoknes et al. [12] and Jasinska et al. [15] reported for good yields of *A. subrufescens*, with the best yield of harvested from C/N 30:1 ($200 \text{ g kg}^{-1}$ and $236 \text{ g kg}^{-1}$ respectively). Both authors have reported that the yield was decreasing with the highest amount of digestate, (lowest C/N 20:1; 161 and $200 \text{ g kg}^{-1}$ respectively [12,15]). On the contrary, in our study the highest yield obtained for Experiment 1 was $156.8 \text{ g kg}^{-1}$, with the lowest C/N ratio 22:1 of the substrate (Tables 3 and 4). Therefore, much more original digestate can be used for *A. subrufescens* cultivation. O'Brien et al. [9] in his study found an inhibiting effect of the use combined diary manure and food waste digestate based on a growing substrate for *Pleurotus ostreatus* cultivation at a C/N ratio lower than 30:1. Furthermore, Zied and Pardo-Giménez [50], stated that *Pleurotus* mushroom cultivation requires a high C:N ratio from 30 up to 300:1. In both studies, high EC (high slat content) of food waste is suggested to be the limiting factor.

Combined diary manure–food waste digestate, like the digestate used in this study, has a high content of protein. Low C/N ratio of digestate makes it perfect for *Agaricus subrufescens*, which prefers a C:N ratio of 10:1 up to as much as 50:1 depending on the source of nitrogen [51]. The low C/N ratio can be beneficial for the mushroom as long as the ammonia is well managed at the end of the second stage of mushroom compost preparation [52].

**Table 4.** Summary of mushroom productivity and cultivation parameters.

| EMC | Bags (Repetitions) | Fresh Weight of Mushrooms from Whole Cropping Period | Yield ($\text{g kg}^{-1}$) | BE (%) | % DM of Mushrooms | Number of Mushrooms (Bag of Substrate) | E (Days) | P (%) |
|---|---|---|---|---|---|---|---|---|
| | | | Experiment 1 | | | | | |
| WD1 | 7 | 944.1 | 44.9 c | 15 c | 13.5 a | 3.6 b | 31.4 a | 68.9 a |
| WD2 | 7 | 2218.7 | 105.7 b | 36 b | 12.7 a | 8.0 a | 29.5 ab | 50.6 ab |
| WD3 | 14 | 4831.5 | 115.0 b | 53 ab | 12.5 a | 9.1 a | 32.5 a | 53.6 ab |
| WD4 | 10 | 4704.9 | 156.8 a | 64 a | 11.8 a | 10.2 a | 25.5 b | 49.5 b |
| | | | Experiment 2a | | | | | |
| WD | 9 | 3227.9 | 119.6 abc | 47 abc | 11.9 a | 9.2 ab | 27.2 a | 51.7 ab |
| O4WD | 5 | 1595.5 | 88.6 c | 38 bc | 9.5 a | 5.8 b | 36.0 a | 57.8 b |
| B4WD | 6 | 1884.6 | 123.2 abc | 48 abc | 11.7 a | 9.0 ab | 26.4 a | 47.7 ab |

**Table 4.** *Cont.*

| EMC | Bags (Repetitions) | Fresh Weight of Mushrooms from Whole Cropping Period | Yield (g kg$^{-1}$) | BE (%) | % DM of Mushrooms | Number of Mushrooms (Bag of Substrate) | E (Days) | P (%) |
|---|---|---|---|---|---|---|---|---|
| | | | Experiment 2b | | | | | |
| O25WD | 5 | 1433.9 | 95.6 bc | 36 c | 10.0 a | 6.6 ab | 34.4 a | 68.9 a |
| B25WD | 5 | 2369.0 | 157.9 a | 64 a | 8.9 a | 10.1 a | 32.2 a | 52.8 ab |
| BOWD | 9 | 3914.9 | 145.0 a | 56 a | 10.5 a | 10.3 a | 29.8 a | 43.3 ab |
| B6WD | 9 | 3605.4 | 133.5 ab | 55 ab | 10.3 a | 10.5 a | 26.1 a | 52.1 ab |

The yield and biological efficiency (BE), earliness (E), and precociousness (P) are defined in the Statistical Analysis section. The column 'Bags' gives the number of bags (units per compost) used for cultivation. Pair-wise significant productivity differences between composts, as determined by the Fisher LSD test, are indicated by lowercase letters; two composts with the same letter are not significantly different (*p* = 0.05). Experimental mushroom composts (EMC): WD—wheat straw–digestate (1,2,3,4-consecutive numbers of EMC); OWD—oat–wheat straw–digestate; BWD—barley–wheat straw digestate; BOWD—oat–barley–wheat straw digestate; the numbers in exp 2a and 2b mean the % of oat or barley straw in substrate).

### 3.2. The Effect Straw Type on Mushroom Productivity and Cultivation Parameters

#### 3.2.1. Mixing and Composting Optimization

The second part of this study focused on the use of different straw types. The substrate was based on wheat straw, which is proven to be the best and the most commonly used for *Agaricus* mushroom cultivation, since it maintains its structure during the composting process [52,53]. However, Norwegian corn production is mostly based on barley, wheat, and oat; thus, there was natural to focus in our research on native corn straws of barley and oat on top of wheat. The addition of barley was from 25% to 60%, whereas oat addition ranged from 15% up to 40% (DM of substrates). Like in Experiment 1, all substrates were aiming for similar initial DM content, ±25–30%. Furthermore, those substrates using straw mixtures were quite easy to work with (like in Experiment 1), although barley straw stalk structure was harder and digestate soaked thorough it; whereas oat straw stalk, softer from origin, soaked with digestate quicker with a tendency to create lumps. Therefore, mixed straw substrates required additional mixing and soaking. In the experiment, we used native varieties: barley "Thermus", oat "Hurdal", and wheat "Mirakel". According to NORSOK [54] Norwegian cereal breeding aims to develop varieties with high yields, strong straw and resistance to fungal diseases and weather extremes, which would explain the difficulties in mixing of substrates.

#### 3.2.2. *Agaricus subrufescens* Productivity and Cultivation Parameters

According to Kopytowski Filho et al. [55] the best C/N ratio for *A. subrufescens* when it is grown in lignocellulosic mushroom composts is of 37/1. The C/N ratio in this study was maintained on rather stable level between 21.9 and 25.2 (Table 3), which did not influence greatly the yield of mushrooms. Therefore, what seemed to have a bigger impact on the mushroom productivity was the addition of different straw types to the original digestate-based cultivation substrate. Comparing all investigated substrates, those with addition of 25% up to 60% barley showed the best productivity in terms of yield (from 157.9 to 123.2 g kg$^{-1}$, respectively) and BE (64% to 48%, respectively) (Table 4) and were in strong positive correlation (Table 5). The barley straw addition tends to have a positive effect on the mushroom productivity; however, yield was slightly decreasing with the growing inclusion (Table 4). As previously mentioned, stalks of barley straw are harder, requiring longer soaking and additional mixing of the substrate mass. However, it also meant that a substrate which has better structure, is less compacted, having more air pockets could be created. Such porous substrate could be colonized faster by the mushroom mycelium and thus obtain a faster yield with mushrooms, which is desired by mushroom growers. The earliness, which represents number of days between the casing and the first harvest, was in fact negatively correlated with yield and biological efficiency (Table 5). Although Fisher LSD method did not show the statistical difference, earliness was the best for barley

straw substrates with higher amount of barley straw (E: 26.1 to 32.2 days on BWD6 to BWD25 respectively) (Tables 2 and 4). Zied et al. [56] also observed negative correlations of earliness and yield and BE. Chankaya et al. [14] showed that the fastest pinning of *Pleurotus* mushrooms was on mixture of corn straw and digestate. For mushroom growers, shorter cultivation cycles mean faster investment return and reduced infestations of flies [57] and contaminants.

Experimental mushroom composts containing oat straw gave lower yield, 88.6 to 95.6 g kg$^{-1}$ and BE of 36% to 38% (Table 4). As mentioned earlier, the oat straw tended to soak very fast with digestate and because of the too soft stalks created lumps. Here, the situation was quite opposite from the barley containing substrates. Substrate could be too compact to create sufficient air conditions for mycelium to grow, and thus resulted in lower yield. This is reflected in the longest period between casing and the first harvest (E = 34 to 36 days, Table 4). However, oat containing substrates had a value for precociousness of 57.8 and 68.9 (Table 4), showing that the production of fruiting bodies accumulated in the first half of the days after first primordia formation, which was also true for study of Zied et al. [56].

Precociousness in Experiment 2 is positively corelated with earliness in all but one substrate (Table 5), and is clearly substrate dependent: the best values for P represent substrates containing oat straw, followed by wheat straw, and barley straw in different amounts. Positive correlation between those two features means that despite the faster pinning on barley, the production of mushrooms accumulated later in the harvest period. This might be a negative feature taking into consideration easier pests and diseases development. On the contrary, for oat containing substrates, precociousness was correlated negatively with time of pinhead formation and mushroom yield and was higher in the first 50 harvesting days. It has been earlier reported that the precociousness values of *A. subrufescens* yields depend on the type of compost and casing used [36]. Other factors influencing precociousness values are the strains used, the compost formulation, substrate texture, physical characteristics of the casing, thickness of the casing layer, and degree of climate control [35,36,58]. However, overall production in our study was better when earliness (time to harvest) was shorter [59,60].

Generally, every cultivation parameter was improved with each batch of compost (Table 4). The number of mushrooms increased (from 3.6 up to 10.2 per bag) and earliness was better (31.4 in the first batch and 25.5 in the last batch). DM of mushrooms however, decreased from 13.5 to 11.8 with no statistical significance. What is interesting is that the mixture of all three straw types in ration 3:1.5:3 (barley: oat: wheat) seems to be the most optimal with high yield and BE (145 g kg$^{-1}$ and 56% respectively), earliness of 29.8 days, and the lowest precociousness of 43.3% (Table 4). Other studies such as Chanakya et al. [14] reported that the mixtures of paddy straw with digestate gave from 20% to 30% better yields for *Pleurotus* ssp. than single substrates (paddy straw alone or digestate alone). Similarly, study of O'Brien et al. [9] (2019) also show, that experimental mushroom composts with more ingredients, here mixture of sawdust, soybean, and combined diary manure and food waste (5:1.5:3.5), tend to perform better in cultivation of *Pleurotus ostreatus*, than use of only sawdust and digestate. The findings above support the assumption that original digestate can be a used directly into mushroom cultivation and thus the additional cost of separation could be omitted. If the mushroom farm could be situated close to the anaerobic digestion plant, sustainable use of assets could be achieved.

**Table 5.** Correlations between mushroom productivity and cultivation parameters in Experiment 2.

| EMC | | Yield | DM | E | P | Number of Mushrooms |
|---|---|---|---|---|---|---|
| WD | DM * | 0.389 | | | | |
| | Earliness | −0.228 | −0.244 | | | |
| | Precociousness | −0.045 | −0.067 | −0.216 | | |
| | Number of mushrooms | 0.267 | 0.267 | −0.527 | −0.414 | |
| | BE ** | 0.500 | 0.389 | −0.228 | −0.045 | 0.267 |
| O4WD | DM * | 0.028 | | | | |
| | Earliness | −0.142 | −0.565 | | | |
| | Precociousness | 0.276 | −0.533 | 0.617 | | |
| | Number of mushrooms | 0.877 | 0.116 | −0.455 | 0.067 | |
| | BE ** | 1.000 | 0.028 | −0.142 | 0.276 | 0.877 |
| B4WD | DM * | −0.578 | | | | |
| | Earliness | −0.712 | 0.965 | | | |
| | Precociousness | −0.049 | 0.218 | 0.019 | | |
| | Number of mushrooms | 0.594 | −0.005 | −0.246 | 0.577 | |
| | BE ** | 1.000 | −0.578 | −0.712 | −0.049 | 0.594 |
| O25WD | DM * | 0.653 | | | | |
| | Earliness | −0.913 | −0.341 | | | |
| | Precociousness | −0.234 | 0.370 | 0.599 | | |
| | Number of mushrooms | 0.556 | 0.159 | −0.687 | −0.478 | |
| | BE ** | 1.000 | 0.653 | −0.913 | −0.234 | 0.556 |
| B25WD | DM * | 0.077 | | | | |
| | Earliness | −0.628 | 0.580 | | | |
| | Precociousness | −0.070 | −0.449 | 0.050 | | |
| | Number of mushrooms | 0.243 | −0.029 | 0.137 | 0.019 | |
| | BE ** | 1.000 | 0.077 | −0.628 | −0.070 | 0.243 |
| BOWD | DM * | −0.379 | | | | |
| | Earliness | −0.063 | −0.222 | | | |
| | Precociousness | 0.413 | −0.445 | 0.432 | | |
| | Number of mushrooms | 0.713 | 0.179 | 0.004 | 0.541 | |
| | BE ** | 1.000 | −0.379 | −0.063 | 0.413 | 0.713 |
| BOWD | DM * | 0.203 | | | | |
| | Earliness | −0.507 | −0.028 | | | |
| | Precociousness | 0.076 | −0.342 | 0.490 | | |
| | Number of mushrooms | 0.679 | 0.639 | −0.255 | −0.308 | |
| | BE ** | 1.000 | 0.203 | −0.507 | 0.076 | 0.679 |

DM *—dry matter of fruiting bodies; BE **—biological efficiency; E—earliness, P—precociousness; experimental mushroom composts (EMC): WD—wheat straw–digestate (1,2,3,4-consecutive numbers of composts); OWD—oat–wheat straw–digestate; BWD—barley–wheat straw digestate; BOWD—oat–barley–wheat straw digestate; the numbers in exp 2a and 2b means the % of oat or barley straw in substrate).



### 4. Conclusions

1.  Using original digestate:
    a.  Mixes well with investigated native corn straws of barley, oat, and wheat;
    b.  Has appropriate moisture of experimental mushroom compost, 64.1% to 78.1% at the point of inoculation, for *A. subrufescens* was achieved without additional watering during composting process;
    c.  Presents good composting process supported substrate colonization and mushroom formation;
    d.  Has appropriate final C/N ratios of experimental mushroom composts, ranging from 21.9 up to 29.7.
2.  Experimental mushroom compost mixes with barley straw were the best performing substrates. Barley enriched substrates showed the best productivity (123.2 up to 157.9 g kg$^{-1}$), BE (48 up to 64%).
3.  Mushroom cultivation parameters such as earliness (time to harvest from casing) was shortest for mixes with barley straw, while precociousness (time of the highest mushroom production after casing) was the lowest for oat mixes.
4.  The optimal digestate mushroom compost for *A. subrufescens* cultivation is a mixture of all three types of native Norwegian straws: barley, oat, and wheat in a 3:1.5:3 ratio.

**Author Contributions:** Conceptualization, A.J.; Methodology, A.J. and K.S.; Validation, K.S.; Formal analysis, E.W.; Investigation, A.J. and M.R.; Resources, A.J. and M.R.; Writing—original draft preparation, A.J.; Writing—review and editing, A.J., K.S. and E.W.; Visualization, A.J.; Supervision, K.S.; Project administration, A.J.; Funding acquisition, A.J. All authors have read and agreed to the published version of the manuscript.

**Funding:** This project has received funding from the European Union's Horizon 2020 research and innovation programme under the Marie Sklodowska-Curie grant agreement no. 751052—VegWaMus CirCrop- H2020-MSCA-IF-2016.

**Institutional Review Board Statement:** Not applicable.

**Informed Consent Statement:** Not applicable.

**Data Availability Statement:** Not applicable.

**Conflicts of Interest:** The authors declare no conflict of interest.

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
