# Peer review of "Bioconversion of Agricultural Wastes into a Value-Added Product: Straw of Norwegian Grains Composted with Dairy Manure Food Waste Digestate in Mushroom Cultivation"

_horticulturae, doi:10.3390/horticulturae8040331_

Round 1

Reviewer 1 Report

Please do not use shortcats in the abstract (BE) -  element symbols are acceptable ;  However, do not use shortcats  such as “barley addition” ( It should be  “barley straw was added” l22). I do not understand particular lines ( for example l. 14 and 15) how you can replace wheat straw with wheat straw. L25 what did you mean by “co-digestion”, L26 circular economics - a fashionable word, but has the closed circuit area  been analyzed? Was it  supported by results and discussed in this respect ????? Is the introduction to circular economics with such general justification justified? I would make a distinction between composting the substrates and preparing the substrate (even though we have aerobic processes in both cases). It would improve the logic of the work. Due to the use of the term compost in the context of the substrate, I believe that the work should be rewritten. Compost is the term for mature material. Isn’t  the title of the paper misleading  suggesting only the preparation of the substrate based on straw and digestate. In tab. 1 only DM is explained, the work is to be understandable also for non-professionals. (it is necessary to refer to the methodology presented below and enter all abbreviations) L202 - according to which method was determined at 120 oC; according to which methods the proportion of water was determined : material for pH and EC

 L214 why 40% and how was organic matter determined?

 L216 does this mean that there were no repetitions!

 L216 if we specify days - unit (l226), then it is appropriate to enter the unit of mass

 L238 Was the normal distribution tested? It is justified to indicate the significance of the test; for me it is unjustified counting  correlation in this experiment. When it comes to  analysis of variance – what is the number of repetitions - I assume it's the number of bags?!. Figures and tables concerning Experimental composts are incomprehensible and illogical. I do not approve of  the use of statistics without checking the normal distribution and reporting, for example, the significance of the correlation coefficient. Inference is too detailed - remove the results or generalize them. The work is multithreaded and this causes that the reader do not understand the main hypothesis. I would divide experiments into separate publications. The authors write about the ease of mixing ……. how was it verified? Because a conclusion is also drawn on this basis?

Author Response

Respected Reviewer, please read attached files for comments response.  

Reviewer 2 Report

The research is well designed and well done. The authors got valuable experimental results which are important both for commercial mushroom production and for agricultural waste utilization. Nevertheless some corrections should be done. They are listed in the notes along the text. Most of them are misprints, necessity to add some missing information in the captions of the tables and recommendation to check attentively the list of cited publications to unify the text. 

The main weakness of the manuscript, to my mind, is the structure of part 3 "Results and discussion". The title of this part means that first the authors should present their own results and then discuss them comparing with literary data. This section is poorly structured. It should be corrected: information related to methods move to appropriate part of methodical section or indicate that optimization of composting methods was a part of the experiments in this study; first present own results, then discuss and compare with other published data. Moreover, subheading 3.1 assumes at least one more subheading - 3.2 - Experiment 2, but it is absent.

All comments are marked in the text of the manuscript.

The manuscript can be accepted for publication after the corrections. 

Author Response

Dear Respected Reviewer, we are thankful to you for your invaluable comments and suggestions for further improvement of our manuscript. The detailed responses to the comments are given in the atached file. 
